# HiGen: Hierarchical Graph Generative Networks

## Abstract

Most real-world graphs exhibit a hierarchical structure, which is often overlooked by existing graph generation methods. To address this limitation, we propose a novel graph generative network that captures the hierarchical nature of graphs and successively generates the graph sub-structures in a coarse-to-fine fashion. At each level of hierarchy, this model generates communities in parallel, followed by the prediction of cross-edges between communities using a separate model. This modular approach results in a highly scalable graph generative network. Moreover, we model the output distribution of edges in the hierarchical graph with a multinomial distribution and derive a recursive factorization for this distribution, enabling us to generate sub-graphs with integer-valued edge weights in an autoregressive approach. Empirical studies demonstrate that the proposed generative model can effectively capture both local and global properties of graphs and achieves state-of-the-art performance in terms of graph quality on various benchmarks.

## 1 Introduction

Graphs play a fundamental role in representing relationships and are widely applicable in various domains. The task of generating graphs from data holds immense value for diverse applications but also poses significant challenges (Dai et al., 2020). Some of the applications include: the exploration of novel molecular and chemical structures (Jin et al., 2020), document generation (Blei et al., 2003), circuit design (Mirhoseini et al., 2021), the analysis and synthesis of realistic data networks, as well as the synthesis of scene graphs in computer (Manolis Savva et al., 2019; Ramakrishnan et al., 2021).

In all the aforementioned domains, a common observation is the presence of locally heterogeneous edge distributions in the graph representing the system, leading to the formation of clusters or communities and hierarchical structures. These clusters represent groups of nodes characterized by a high density of edges within the group and a comparatively lower density of edges connecting the group with the rest of the graph. In a hierarchical structure that arise from graph clustering, the communities in the lower levels capture the local structures and relationships within the graph. These communities provide insights into the fine-grained interactions among nodes. On the other hand, the higher levels of the hierarchy reflect the broader interactions between communities and characterize global properties of the graph. Therefore, in order to generate realistic graphs, it is essential for graph generation models to learn this multi-scale structure, and be able to capture the cross-level relations. While hierarchical multi-resolution generative models were developed for specific data types such as voice (Oord et al., 2016), image (Reed et al., 2017; Karami et al., 2019) and molecular motifs Jin et al. (2020), these methods rely on domain-specific priors that are not applicable to general graphs with unordered nature. To the best of our knowledge, there exists no data-driven generative models specifically designed for generic graphs that can effectively incorporate hierarchical structure.

Graph generative models have been extensively studied in the literature. Classical methods based on random graph theory, such as those proposed in Erdos & Rényi (1960) and Barabási & Albert (1999),

can only capture a limited set of hand-engineered graph statistics. Leskovec et al. (2010) leveraged the Kronecker product of matrices but the resulting generative model is very limited in modeling the underlying graph distributions. With recent advances in graph neural networks, a variety of deep neural network models have been introduced that are based on variational autoencoders (VAE) (Kingma & Welling, 2013) or generative adversarial networks (GAN) (Goodfellow et al., 2020). Some examples of such models include (De Cao & Kipf, 2018; Simonovsky & Komodakis, 2018; Kipf & Welling, 2016; Ma et al., 2018; Liu et al., 2019; Bojchevski et al., 2018; Yang et al., 2019) The major challenge in VAE based models is that they rely on heuristics to solve a graph matching problem for aligning the VAE's input and sampled output, limiting them to small graphs. On the other hand, GAN-based methods circumvent the need for graph matching by using a permutation invariant discriminator. However, they can still suffer from convergence issues and have difficulty capturing complex dependencies in graph structures for moderate to large graphs Li et al. (2018); Martinkus et al. (2022). To address these limitations, (Martinkus et al., 2022) recently proposed using spectral conditioning to enhance the expressivity of GAN models in capturing global graph properties.

On the other hand, autoregressive models approach graph generation as a sequential decision-making process. Following this paradigm, Li et al. (2018) proposed generative model based on GNN but it has high complexity of $\mathcal{O}(mn^2)$. In a distinct approach, GraphRNN (You et al. 2018) modeled graph generation with a two-stage RNN architecture for generating new nodes and their links, respectively. However, traversing all elements of the adjacency matrix in a predefined order results in $\mathcal{O}(n^2)$ time complexity making it non-scalable to large graphs. In contrast, GRAN (Liao et al., 2019) employs a graph attention network and generates the adjacency matrix row by row, resulting in a $\mathcal{O}(n)$ complexity sequential generation process. To improve the scalability of generative models, Dai et al. (2020) proposed an algorithm for sparse graphs that decreases the training complexity to $\mathcal{O}(\log n)$, but at the expense of increasing the generation time complexity to $\mathcal{O}((n+m)\log n)$. Despite their improvement in capturing complex statistics of the graphs, autoregressive models highly rely on an appropriate node ordering and do not take into account the community structures of the graphs. Additionally, due to their recursive nature, they are not fully parallelizable.

A new family of diffusion model for graphs has emerged recently. Continuous denoising diffusion was developed by Jo et al. (2022), which adds Gaussian noise to the graph adjacency matrix and node features during the diffusion process. However, since continuous noise destroys the sparsity and structural properties of the graph, discrete denoising diffusion models have been developed as a solution in (Haefeli et al., 2022; Vignac et al., 2022). These models progressively edit graphs by adding or removing edges in the diffusion process, and then denoising graph neural networks are trained to reverse the diffusion process. While the denoising diffusion models can offer promising results, their main drawback is the requirement of a long chain of reverse diffusion, which can result in relatively slow sampling.

In his work, we introduce HiGen, a **Hi**erarchical **G**raph Ge**n**erative Network to address the limitations of existing generative models by incorporating community structures and cross-level interactions. This approach involves generating graphs in a coarse-to-fine manner, where graph generation at each level is conditioned on a higher level (lower resolution) graph. The generation of communities at lower levels is performed in parallel, followed by the prediction of cross-edges between communities using a separate model. This parallelized approach enables high scalability. To capture hierarchical relations, our model allows each node at a given level to depend not only on its neighbouring nodes but also on its corresponding super-node at the higher level. Furthermore, we address the generation of integer-valued edge weights of the hierarchical structure by modeling the output distribution of edges using a multinomial distribution. We show that multinomial distribution can be factorized successively, enabling the autoregressive generation of each community. This property makes the proposed architecture well-suited for generating graphs with integer-valued edge weights. Furthermore, by breaking down the graph generation process into the generation of multiple small partitions that are conditionally independent of each other, HiGen reduces its sensitivity to a predefined initial ordering of nodes.

## 2 Background

A graph $\mathcal{G} = (\mathcal{V}, \mathcal{E})$ is a collection of nodes (vertices) $\mathcal{V}$ and edges $\mathcal{E}$ with corresponding sizes $n = |\mathcal{V}|$ and $m = |\mathcal{E}|$ and an adjacency matrix $\mathbf{A}^\pi$ for the node ordering $\pi$. The node set can be partitioned into $c$ communities (a.k.a. cluster or modules) using a graph partitioning function

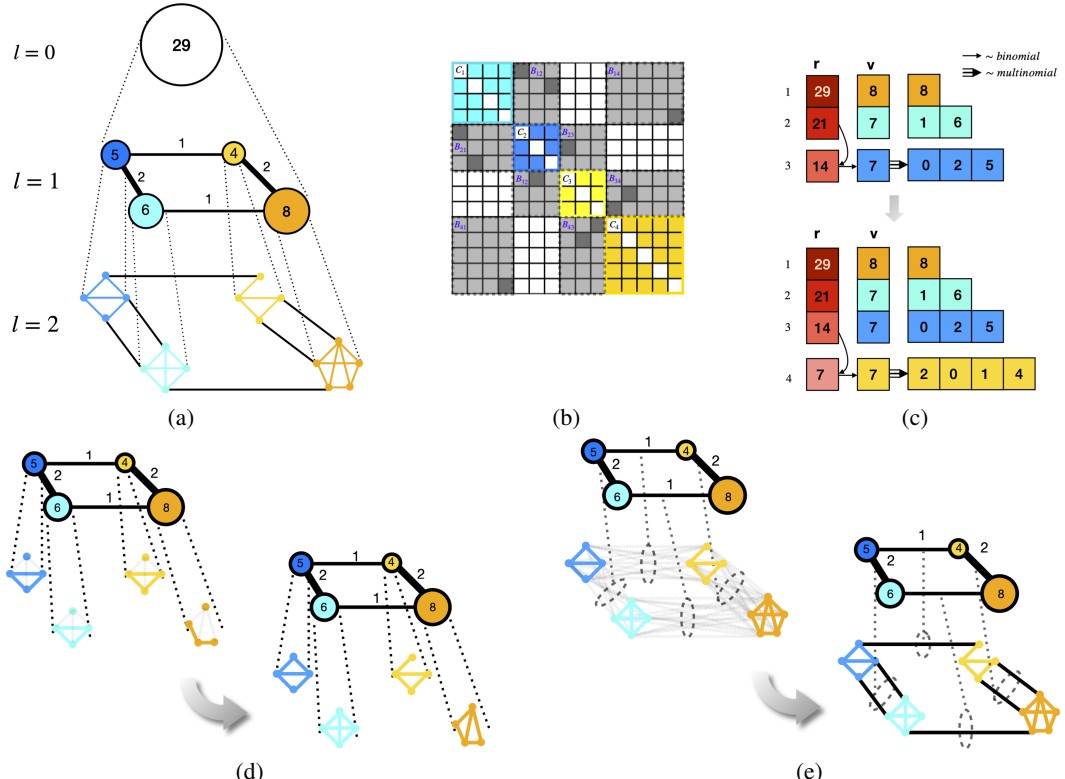

Figure 1: (a) A sample hierarchical graph with 2 levels is shown. Communities are shown in different colors and the weight of a node and the weight of an edge in a higher level, represent the sum of the edges in the corresponding community and bipartite, respectively. Node size and edge width indicate their weights. (b) The matrix shows corresponding adjacency of the graph $\mathcal{G}^2$ matrix where each of its sub-graphs corresponds to a block in the adjacency matrix, communities are shown in different colors and bipartites are colored in gray. (c) Decomposition of multinomial distribution as a recursive *stick-breaking* process where at each iteration, first a fraction of the remaining weights $\mathrm{w}_m$ is allocated to the $m$-th row (the $m$-th node in the sub-graph) and then this fraction $\mathrm{v}_m$ is distributed among that row of lower triangular adjacency matrix, $\hat{A}$. (d) Parallel generation of communities. (e) Parallel prediction of bipartites. Shadowed lines are the *augmented edges* representing candidate edges at each step.

$\mathcal{F}: \mathcal{V} \to \{1, ..., c\}$, where each cluster of nodes forms a sub-graph denoted by $\mathcal{C}_i = (\mathcal{V}(\mathcal{C}_i), \mathcal{E}(\mathcal{C}_i))$ with adjacency matrix $\mathbf{A}_i$. The cross-links between neighboring communities form a *bipartite graph*, denoted by $\mathcal{B}_{ij} = (\mathcal{V}(\mathcal{C}_i), \mathcal{V}(\mathcal{C}_j), \mathcal{E}(\mathcal{B}_{ij}))$ with adjacency matrix $\mathbf{A}_{ij}$. Each community is aggregated to a super-node and each bipartite corresponds to a super-edge linking neighboring communities, which induces a coarser graph at the higher (a.k.a. parent) level. Herein, the levels are indexed by superscripts. Formally, each community at level $l$, $\mathcal{C}_i^l$, is mapped to a node at the higher level graph, also called its parent node, $v_i^{l-1} := Pa(\mathcal{C}_i^l)$ and each bipartite at level $l$ is represented by an edge in the higher level, also called its parent edge, $e_i^{l-1} = Pa(\mathcal{B}_{ij}^l) = (v_i^{l-1}, v_j^{l-1})$. The weights of the self edges and the weights of the cross-edges in the parent level are determined by the sum of the weights of the edges within their corresponding community and bipartite, respectively. Therefore, the edges in the induced graphs at the higher levels have integer-valued weights: $w_{ii}^{l-1} = \sum_{e \in \mathcal{E}(\mathcal{C}_i^l)} w_e$ and $w_{ij}^{l-1} = \sum_{e \in \mathcal{E}(\mathcal{B}_{ij}^l)} w_e$, moreover sum of all edge weights remains constant in all levels so $w_0 := \sum_{e \in \mathcal{E}(\mathcal{G}^l)} w_e = |\mathcal{E}|, \ \forall \ l \in [0, ..., L]$.

This clustering process continues recursively in a bottom-up approach until a single node graph $\mathcal{G}^0$ is obtained, producing a *hierarchical graph*, defined by the set of graphs in all levels of abstractions, $\mathcal{HG} := \{\mathcal{G}^0, ...., \mathcal{G}^{L-1}, \mathcal{G}^L\}$. This forms a dendrogram tree with $\mathcal{G}^0$ being the root and $\mathcal{G}^L$ being the final graph that is generated at the leaf level. An $\mathcal{HG}$ is visualized in figure 1a. The hierarchical tree

structure enables modeling of both local and long-range interactions among nodes, as well as control over the flow of information between them, across multiple levels of abstraction. This is a key aspect of our proposed generative model.

## 3 Hierarchical Graph Generation

In graph generative networks, the objective is to learn a generative model, $p(\mathcal{G})$ given a set of training graphs. This work aims to establish a hierarchical multi-resolution framework for generating graphs in a coarse-to-fine fashion. In this framework, we assume that the graphs do not have node attributes, so the generative model only needs to characterize the graph topology. Given a particular node ordering $\pi$, and a hierarchical graph $\mathcal{HG} := \{\mathcal{G}^0, ...., \mathcal{G}^{L-1}, \mathcal{G}^L\}$, produced by recursively applying a graph partitioning function, $\mathcal{F}$, we can factorize the generative model using the chain rule of probability as:

$$p(\mathcal{G} = \mathcal{G}^L, \pi) = p(\{\mathcal{G}^L, \mathcal{G}^{L-1}, ..., \mathcal{G}^0\}, \pi) = p(\mathcal{G}^L, \pi \mid \{\mathcal{G}^{L-1}, ..., \mathcal{G}^0\}) \, ... \, p(\mathcal{G}^1, \pi \mid \mathcal{G}^0) \, p(\mathcal{G}^0)$$

$$= \prod_{l=0}^{L} p(\mathcal{G}^l, \pi \mid \mathcal{G}^{l-1}) \times p(\mathcal{G}^0) \tag{1}$$

In other words, the generative process involves specifying the probability of the graph at each level conditioned on its parent level graph in the hierarchy. This process is iterated recursively until the lowest level, or leaf level, is reached. Here, the distribution of the root $p(\mathcal{G}^0) = p(\mathrm{w}^0 = w_0)$ can be simply estimated using the empirical distribution of the number of edges $|\mathcal{E}|$ of graphs in the training set.

Based on the partitioned structure within each level of $\mathcal{HG}$, the conditional generative probability $p(\mathcal{G}^l \mid \mathcal{G}^{l-1})$ can be decomposed into the probability of its communities and bipartite graphs as:

$$p(\mathcal{G}^l \mid \mathcal{G}^{l-1}) = p(\{\mathcal{C}_i^l \ \forall i \in \mathcal{V}(\mathcal{G}^{l-1})\} \ \cup \ \{\mathcal{B}_{ij}^l \ \forall (i,j) \in \mathcal{E}(\mathcal{G}^{l-1})\} \mid \mathcal{G}^{l-1})$$

$$\eqsim \prod_{i \, \in \, \mathcal{V}(\mathcal{G}^{l-1})} p(\mathcal{C}_i^l \mid \mathcal{G}^{l-1}) \times \prod_{(i,j) \in \, \mathcal{E}(\mathcal{G}^{l-1})} p(\mathcal{B}_{ij}^l \mid \mathcal{G}^{l-1}) \tag{2}$$

The approximation in this decomposition becomes an equivalence when each community $\mathcal{C}_i^l$ or bipartite graph $\mathcal{B}_{ij}^l$ is assumed to be independent of all other components in its level conditioned on the parent graph $\mathcal{G}^{l-1}$.[1] Since the integer-valued weights of the edges in each level can be modeled by a multinomial distribution, we can leverage the properties of multinomial distribution to prove the conditional independence of the components.

**Theorem 3.1.** *Let the random vector* $\mathbf{w} := [w_e]_{e \, \in \, \mathcal{E}(\mathcal{G}^l)}$ *denote the set of weights of all edges of* $\mathcal{G}^l$ *such that their sum is* $w_0 = \mathbf{1}^T \mathbf{w}$. *The joint probability of* $\mathbf{w}$ *can be described by a multinomial distribution:* $\mathbf{w} \sim Mu(\mathbf{w} \mid w_0, \boldsymbol{\theta}^l)$. *By observing that the sum of edge weights within each community* $\mathcal{C}_i^l$ *and bipartite graph* $\mathcal{B}_{ij}^l$ *are determined by the weights of their parent edges in the higher level,* $w_{ii}^{l-1}$ *and* $w_{ij}^{l-1}$ *respectively, we can establish that these components are conditionally independent and each of them follow a multinomial distribution:*

$$p(\mathcal{G}^l \mid \mathcal{G}^{l-1}) \sim \prod_{i \, \in \, \mathcal{V}(\mathcal{G}^{l-1})} Mu([w_e]_{e \, \in \, \mathcal{C}_i^l} \mid w_{ii}^{l-1}, \boldsymbol{\theta}_{ii}^l) \times \prod_{(i,j) \in \, \mathcal{E}(\mathcal{G}^{l-1})} Mu([w_e]_{e \, \in \, \mathcal{B}_{ij}^l} \mid w_{ij}^{l-1}, \boldsymbol{\theta}_{ij}^l)$$

$$\tag{3}$$

*where* $\{\boldsymbol{\theta}_{ij}^l[e] \in [0,1], \ s.t. \ \mathbf{1}^T \boldsymbol{\theta}_{ij}^l = 1 \mid \forall \, (i,j) \in \ \mathcal{E}(\mathcal{G}^{l-1})\}$ *are the multinomial model parameters.*

*Proof.* The detailed proof can be found in Appendix A.1 □

Therefore, given the parent graph at a higher level, the generation of graph at its subsequent level can be reduced to generation of its partition and bipartite sub-graphs. As illustrated in figure, this decomposition enables parallel generation of the communities in each level which can be followed by predicting all bipartite sub-graphs in that level at one pass. Each of these sub-graphs corresponds to a block in the adjacency matrix, as visualized in figure 1b, so the proposed hierarchical model generates adjacency matrix in a blocks-wise fashion and constructs the final graph topology.

---

[1]Indeed, this assumption implies that the cross dependency between communities are primarily encoded by their parent abstract graph which is reasonable where the nodes' dependencies are mostly local and are within community rather than being global.

## 3.1 Community Generation

Based on the equation (3), the edge weights within each community can be jointly modeled using a multinomial distribution. Our objective is to model the generative probability of communities in each level as an autoregressive process. To accomplish this, we need to factorize the multinomial distribution accordingly. Toward this goal, we present two different approaches in the following.

**Lemma 3.2.** *A random counting vector* $\mathbf{w} \in \mathbb{Z}_+^E$ *with a multinomial distribution can be recursively decomposed into a sequence of binomial distributions as follows:*

$$Mu(\mathbf{w}_1, ..., \mathbf{w}_E \mid w, [\theta_1, ..., \theta_E]) = \prod_{e=1}^{E} Bi(\mathbf{w}_e \mid w - \sum\nolimits_{i<e} \mathbf{w}_i, \hat{\theta}_e), \qquad (4)$$

$$where: \hat{\theta}_e = \frac{\theta_e}{1 - \sum_{i<e} \theta_i}$$

*This decomposition is known as a stick-breaking process, where* $\hat{\theta}_e$ *is the fraction of the remaining probabilities we take away every time and allocate to the* $e$*-th component (Linderman et al., 2015).*

This lemma enable us to model the generation of a community as an edge-by-edge autoregressive process, similar to existing algorithms such as GraphRNN (You et al., 2018) or DeepGMG (Li et al., 2018) with $\mathcal{O}(|\mathcal{V}_{\mathcal{C}}|^2)$ generation steps. However, inspired by GRAN (Liao et al., 2019), a community can be generated more efficiently by generating one node at a time. This requires decomposing the generative probability of edges in a group-wise form, where the candidate edges between the $t$-th node and the already generated graph are grouped together. In other words, this model completes the lower triangle adjacency matrix one row at a time conditioned on the already generated sub-graph and the parent-level graph. The following theorem formally derives this decomposition for multinomial distributions.

**Theorem 3.3.** *For a random counting vector* $\mathbf{w} \in \mathbb{Z}_+^E$ *with a multinomial distribution* $Mu(\mathbf{w} \mid w, \boldsymbol{\theta})$, *let's split it into* $M$ *disjoint groups* $\mathbf{w} = [\mathbf{u}_1, ..., \mathbf{u}_M]$ *where* $\mathbf{u}_m \in \mathbb{Z}_+^{E_m}$, $\sum_{m=1}^{M} E_m = E$, *and also split the probability vector accordingly as* $\boldsymbol{\theta} = [\boldsymbol{\theta}_1, ..., \boldsymbol{\theta}_M]$. *Additionally, let's define sum of all variables in the* $m$*-th group by a random count variable* $\mathbf{v}_m := \sum_{e=1}^{E_m} \mathbf{u}_{m,e}$. *Then, the multinomial distribution can be factorized as a chain of binomial and multinomial distributions:*

$$Mu(\mathbf{w} = [\mathbf{u}_1, ..., \mathbf{u}_M] \mid w, \boldsymbol{\theta} = [\boldsymbol{\theta}_1, ..., \boldsymbol{\theta}_M]) = \prod_{m=1}^{M} Bi(\mathbf{v}_m \mid w - \sum_{i<m} \mathbf{v}_i, \eta_{\mathbf{v}_m}) \, Mu(\mathbf{u}_m \mid \mathbf{v}_m, \boldsymbol{\lambda}_m),$$

$$where: \eta_{\mathbf{v}_m} = \frac{\mathbf{1}^T \boldsymbol{\theta}_m}{1 - \sum_{i<m} \mathbf{1}^T \boldsymbol{\theta}_i}, \quad \boldsymbol{\lambda}_m = \frac{\boldsymbol{\theta}_m}{\mathbf{1}^T \boldsymbol{\theta}_m}. \qquad (5)$$

*Here, the probability of binomial,* $\eta_{\mathbf{v}_m}$, *is the fraction of the remaining probability mass that is allocated to* $\mathbf{v}_m$, *i.e. the sum of all weights in the* $m$*-th group. The vector parameter* $\boldsymbol{\lambda}_m$ *is the normalized multinomial probabilities of all count variables in the* $m$*-th group. Intuitively, this decomposition of multinomial distribution can be viewed as a recursive stick-breaking process where at each step, first a binomial distribution is used to determine how much probability mass to allocate to the current group, and a multinomial distribution is used to distribute that probability mass among the variables in the group. The resulting distribution is equivalent to the original multinomial distribution.*

*Proof.* Refer to appendix A.2 for the proof. □

Let $\hat{\mathcal{C}}_{i,t}^l$ denote an already generated sub-graph, at the $t$-th step, augmented with the set of candidate edges, from the new node, $v_t(\mathcal{C}_i^l)$, to its preceding node denoted by $\hat{\mathcal{E}}_t(\hat{\mathcal{C}}_{i,t}^l) := \{(t, j) \mid j < t\}$. We collect the weights of these edges in the random vector $\mathbf{u}_t := [w_e]_{e \in \hat{\mathcal{E}}_t(\hat{\mathcal{C}}_{i,t}^l)}$ (that is the $t$-th row of the lower triangle of adjacency matrix $\hat{\mathbf{A}}_i^l$), where the sum of the candidate edge weights is $\mathbf{v}_t$. Based on theorem 3.3, the probability of $\mathbf{u}_t$ can be characterized by the product of a binomial and a multinomial distribution. This process is illustrated in figure 1c. We further increase the

expressiveness of the generative network by extending this probability to a mixture model with $K$ mixtures:

$$p(\mathbf{u}_t) = \sum_{k=1}^{K} \boldsymbol{\beta}_k^l \mathrm{Bi}(\mathrm{v}_t | w_{ii}^{l-1} - \sum_{i<t} \mathrm{v}_i, \eta_{t,k}^l) \mathrm{Mu}(\mathbf{u}_t \,|\, \mathrm{v}_t, \boldsymbol{\lambda}_{t,k}^l) \tag{6}$$

$$\boldsymbol{\lambda}_{t,k}^l = \mathrm{softmax}\left(\mathrm{MLP}_{\boldsymbol{\theta}}^l\big(\left[\Delta\boldsymbol{h}_{\hat{\mathcal{E}}_t(\hat{\mathcal{C}}_{i,t}^l)} \,||\, h_{Pa(\mathcal{C}_i^l)}\right]\big)\right)[k,:] \tag{7}$$

$$\eta_{t,k}^l = \mathrm{sigmoid}\left(\mathrm{MLP}_{\eta}^l\big(\left[\mathrm{pool}(\boldsymbol{h}_{\hat{\mathcal{C}}_{i,t}^l}) \,||\, h_{Pa(\mathcal{C}_i^l)}\right]\big)\right)[k]$$

$$\boldsymbol{\beta}^l = \mathrm{softmax}\left(\mathrm{MLP}_{\beta}^l\big(\left[\mathrm{pool}(\boldsymbol{h}_{\hat{\mathcal{C}}_{i,t}^l}) \,||\, h_{Pa(\mathcal{C}_i^l)}\right]\big)\right)$$

Where $\Delta\boldsymbol{h}_{\hat{\mathcal{E}}_t(\hat{\mathcal{C}}_{i,t}^l)}$ is a $|\hat{\mathcal{E}}_t(\hat{\mathcal{C}}_{i,t}^l)| \times d_h$ dimensional matrix, consisting of the set of edge features $\{\Delta h_{(t,s)} := h_t - h_s \mid \forall \, (t,s) \in \hat{\mathcal{E}}_t(\hat{\mathcal{C}}_{i,t}^l)\}$, $\boldsymbol{h}_{\hat{\mathcal{C}}_{i,t}^l}$ is a $t \times d_h$ matrix of node features in the augmented community graph. The mixture weights are denoted by $\boldsymbol{\beta}^l$. Here, the node features are learned by GNN models and the graph level representation is obtained by the $\mathrm{addpool}()$ aggregation function. In order to produce $K \times |\mathcal{E}_t(\mathcal{C}_i^l)|$ dimensional matrix of multinomial probabilities, the $\mathrm{MLP}_{\boldsymbol{\theta}}^l()$ network acts at the edge level, while $\mathrm{MLP}_{\eta\mathrm{v}}^l()$ and $\mathrm{MLP}_{\beta}^l()$ act at the graph level to produce the binomial probabilities and $K$ dimensional arrays for $K$ mixture models, respectively. All of these MLP networks are built by two hidden layers with $\mathrm{ReLU}()$ activation functions.

During the generation process of each community $\mathcal{C}_i^l$, the node features of its parent node $h_{Pa(\mathcal{C}_i^l)}$ are used as the context. This context is concatenated to the node and edge feature matrices using the operation $\left[\boldsymbol{x} \,||\, y\right]$, which concatenates vector $y$ to each row of matrix $\boldsymbol{x}$. The purpose of this context is to enrich the node and edge features by capturing long-range interactions and encoding the global structure of the graph, which is important for generating local components.

## 3.2  Bipartite Generation

Once all the communities in level $l$ are generated, the edges of all bipartite graphs at that level can be predicted simultaneously. An augmented graph $\hat{\mathcal{G}}^l$ composed of all the communities, $\{\mathcal{C}_i^l \ \forall i \in \mathcal{V}(\mathcal{G}^{l-1})\}$, and the candidate edges of all bipartites, $\{\mathcal{B}_{ij}^l \ \forall(i,j) \in \mathcal{E}(\mathcal{G}^{l-1})\}$, is used as the input of a GNN to obtain node and edge features. We similarly extend the multinomial distribution of a bipartite, (12), using a mixture model to express its generative probability:

$$p(\mathbf{w} := \hat{\mathcal{E}}(\mathcal{B}_{ij}^l)) = \sum_{k=1}^{K} \boldsymbol{\beta}_k^l \mathrm{Mu}(\mathbf{w} \mid w_{ij}^{l-1}, \boldsymbol{\theta}_{ij,k}^l)$$

$$\boldsymbol{\theta}_{ij,k}^l = \mathrm{softmax}\left(\mathrm{MLP}_{\boldsymbol{\theta}}^l(\left[\Delta\boldsymbol{h}_{\hat{\mathcal{E}}(\mathcal{B}_{ij})} \,||\, \Delta h_{Pa(\mathcal{B}_{ij})}\right])\right)[k,:] \tag{8}$$

$$\boldsymbol{\beta}^l = \mathrm{softmax}\left(\mathrm{MLP}_{\beta}^l\big(\left[\mathrm{pool}(\Delta\boldsymbol{h}_{\hat{\mathcal{E}}(\mathcal{B}_{ij})}) \,||\, \Delta h_{Pa(\mathcal{B}_{ij})}\right]\big)\right)$$

where the random vector $\mathbf{w} := [w_e]_{e \,\in\, \hat{\mathcal{E}}(\mathcal{B}_{ij}^l)}$ is the set of weights of all candidate edges in bipartite $\mathcal{B}_{ij}^l$ and $\Delta\boldsymbol{h}_{Pa(\mathcal{B}_{ij}^l)}$ are the parent edge features of the bipartite graph.

**Node Feature Encoding:**  To encode node features, we extend GraphGPS proposed by Rampášek et al. (2022). GraphGPS combines local message-passing with global attention mechanism and uses positional and structural encoding for nodes and edges to construct a more expressive and a scalable graph transformer (GT) (Dwivedi & Bresson, 2020). To apply GraphGPS on augmented graphs, we use distinct initial edge features to distinguish augmented (candidate) edges from real edges. Furthermore, for bipartite generation, the attention scores in the Transformers of the augmented graph $\hat{\mathcal{G}}^l$ are masked to restrict attention only to connected communities. The details of model architecture are provided in appendix B.

## 4 Related Work

In order to deal with hierarchical structures in molecular graphs, a generative process was proposed by Jin et al. (2020) which recursively selects motifs, the basic building blocks, from a set and predicts their attachment to the emerging molecule. However, this method requires prior domain-specific knowledge and relies on molecule-specific graph motifs. Additionally, the graphs are only abstracted into two levels, and component generation cannot be performed in parallel. In (Kuznetsov & Polykovskiy, 2021), a hierarchical normalizing flow model for molecular graphs was introduced, where new molecules are generated from a single node by recursively dividing each node into two. However, the merging and splitting of pairs of nodes in this model is based on the node's neighborhood, and do not consider the diverse community structure of graphs, therefore the hierarchical generation of this model is inherently limited.

## 5 Experiments

In our empirical studies, we compare the proposed hierarchical graph generative network against state-of-the-art autoregressive models: GRAN and GraphRNN models, diffusion models: DiGress (Vignac et al., 2022) and GDSS (Jo et al., 2022) and a GAN-based model: SPECTRE (Martinkus et al., 2022), on a range of synthetics and real datasets of various sizes.

**Datasets:** We used 4 different benchmark graph datasets: (1) the synthetic *Stochastic Block Model (SBM)* dataset consisting of 200 graphs with 2-5 communities each with 20-40 nodes, used in a previous work (Martinkus et al., 2022); (2) the *Protein* including 918 protein graphs, each has 100 to 500 nodes representing amino acids that are linked if they are closer than 6 Angstroms (Dobson & Doig, 2003), (3) the *Enzyme* that has 587 protein graphs of 10-125 nodes, representing protein tertiary structures of the enzymes from the BRENDA database (Schomburg et al., 2004) and (4) the *Ego* dataset containing 757 3-hop ego networks with 50-300 nodes extracted from the CiteSeer dataset, where nodes represent documents and edges represent citation relationships (Sen et al., 2008).

**Graph Partitioning** Different algorithms approach the problem of graph partitioning (clustering) using various clustering quality functions. Two commonly used families of such metrics are modularity and cut-based metrics (Tsitsulin et al., 2020). Although optimizing modularity metric is an NP-hard problem, it is well-studied in the literature and several graph partitioning algorithm based on this metric have been proposed. For example, the Louvain algorithm (Blondel et al., 2008) starts with each node as its community and then repeatedly merges communities based on the highest increase in modularity until no further improvement can be made. This heuristic algorithm is computationally efficient and scalable to large graphs for community detection. Moreover, a spectral relaxation of modularity metrics has been proposed in Newman (2006a;b) which results in an analytically solution for graph partitioning. Additionally, an unsupervised GNN-based pooling method inspired by this spectral relaxation was proposed for partitioning graphs with node attributes (Tsitsulin et al., 2020). As the modularity metric is based on the graph structure, it is well-suited for our problem. Therefore, we employed the Louvain algorithm to hierarchically cluster the graph datasets in our experiments and then spliced out the intermediate levels to achieve HGs with uniform depth of $L = 2$.

**Model Architecture** In the experiments, the GNN models consist of 8 layers of GraphGPS layers (Rampášek et al., 2022). The input node feature of GNNs is augmented with positional and structural encoding, where the first 8 eigenvectors corresponding to the smallest non-zero eigenvalues of the Laplacian and diagonal of the random-walk matrix up to 8-steps are used. Each level has its own GNN and output models. The details of the model architecture are presented in Appendix B and C.

We conducted experiments using the proposed hierarchical graph generative network (HiGen) model with two variants for the output distribution of the leaf edges: 1) **HiGen**: the probability of the community edges' weights at the leaf level are modeled by mixture of Bernoulli, using sigmoid() activation in equation 7, since the leaf levels in our experiments have binary edges weights, while higher levels use mixture of multinomials. 2)**HiGen-m**: the model uses a mixture of multinomial distributions (6) to describe the output distribution for all levels. In this case, we observed that modeling the probability parameters of edge weights of the leaf level, denoted as $\boldsymbol{\lambda}_{t,k}$ in (7), by a multi-hot activation function, defined as $\sigma(\mathbf{z})_i := \text{sigmoid}(z_i)/\sum_{j=1}^{K} \text{sigmoid}(z_j)$ where $\sigma : \mathbb{R}^K \to (K-1)$-simplex, provided slightly better performance than the standard softmax() function. However, for

Table 1: Comparison of generation metrics on benchmark datasets. The baseline results for SBM and Protein graphs are obtained from (Martinkus et al., 2022; Vignac et al., 2022), and the results for enzyme graphs (except for GRAN, which we implemented) are obtained from (Jo et al., 2022), while we implemented them for Ego. "-": not applicable due to resource issue or not reported in the reference papers.

| Model | Stochastic block model | | | | Protein | | | |
|---|---|---|---|---|---|---|---|---|
| | Deg. ↓ | Clus. ↓ | Orbit↓ | Spec.↓ | Deg. ↓ | Clus. ↓ | Orbit↓ | Spec.↓ |
| Training set | 0.0008 | 0.0332 | 0.0255 | 0.0063 | 0.0003 | 0.0068 | 0.0032 | 0.0009 |
| GraphRNN | 0.0055 | 0.0584 | 0.0785 | 0.0065 | 0.0040 | 0.1475 | 0.5851 | 0.0152 |
| GRAN | 0.0113 | 0.0553 | 0.0540 | 0.0054 | 0.0479 | 0.1234 | 0.3458 | 0.0125 |
| SPECTRE | 0.0015 | 0.0521 | 0.0412 | 0.0056 | 0.0056 | 0.0843 | 0.0267 | 0.0052 |
| DiGress | **0.0013** | **0.0498** | 0.0433 | - | - | - | - | - |
| HiGen-m | 0.0017 | 0.0503 | 0.0604 | 0.0068 | 0.0041 | 0.109 | 0.0472 | 0.0061 |
| HiGen | 0.0019 | **0.0498** | 0.0352 | **0.0046** | **0.0012** | **0.0435** | **0.0234** | **0.0025** |

| Model | Enzyme | | | Ego | | | |
|---|---|---|---|---|---|---|---|
| | Deg. ↓ | Clus. ↓ | Orbit ↓ | Deg. ↓ | Clus. ↓ | Orbit ↓ | Spec. ↓ |
| Training set | 0.0011 | 0.0025 | 3.7e-4 | 2.2e-4 | 0.010 | 0.012 | 1.4e-3 |
| GraphRNN | 0.017 | 0.062 | 0.046 | 0.024 | 0.34 | 0.14 | 0.089 |
| GRAN | 0.054 | 0.087 | 0.033 | 0.032 | 0.17 | 0.026 | 0.046 |
| GDSS | 0.026 | 0.061 | 0.009 | - | - | - | - |
| HiGen-m | 0.027 | 0.157 | 1.2e-3 | 0.011 | 0.063 | **0.021** | 0.013 |
| HiGen | **0.012** | **0.038** | **7.2**e-4 | **1.9**e-3 | **0.049** | 0.029 | **0.004** |

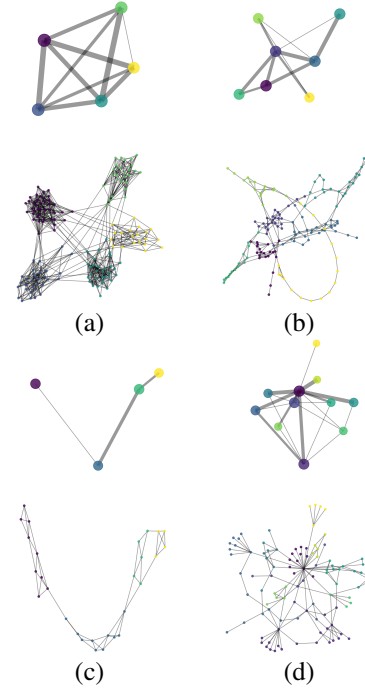

(a)   (b)

(c)   (d)

Figure 2: Samples from HiGen. a) SBM, b) Protein, c) Enzyme and d) Ego. Communities are distinguished with different colors and both levels are depicted.

both HiGen and HiGen-m, the probabilities of the integer-valued edges at the higher levels are still modeled by the standard softmax() function.[2]

For training, HiGen models used the Adam optimizer Kingma & Ba (2014) with a learning rate of 5e-4 and its default settings of $\beta_1 = 0.9$, $\beta_2 = 0.999$ and $\epsilon$=1e-8.

**Metrics**   To evaluate the graph generative models, we adopt the approach proposed in (Liu et al., 2019; Liao et al., 2019), which compares the distributions of four different graph statistics between the ground truth and generated graphs: (1) degree distributions, (2) clustering coefficient distributions, (3) the number of occurrences of all orbits with four nodes, and (4) the spectra of the graphs by computing the eigenvalues of the normalized graph Laplacian. The first three metrics capture local graph statistics, while the spectra represents global structure. The maximum mean discrepancy (MMD) score over these statistics are used as the metrics. While Liu et al. (2019) computed MMD scores using the computationally expensive Gaussian earth mover's distance (EMD) kernel, Liao et al. (2019) proposed using the total variation (TV) distance as an alternative measure. TV distance is much faster and still consistent with the Gaussian EMD kernel. Most recently, O'Bray et al. (2021) suggested using other efficient kernels such as an RBF kernel, or a Laplacian kernel, or a linear kernel. Additionally, Thompson et al. (2022) proposed new evaluation metrics for comparing graph sets using a random-GNN approach where GNNs are employed to extract meaningful graph features. However, in this work, we follow the experimental setup and evaluation metrics of (Liao et al., 2019), except for the enzyme dataset where we use a Gaussian EMD kernel to be consistent with the results reported in (Jo et al., 2022). GNN-based performance metrics of HiGen model are also reported in appendix D.2.

---

[2]As the leaf levels have binary edge weights while the sum of their weights is determined by their parent edge, a possible extension to this work could be using the cardinality potential model (Hajimirsadeghi et al., 2015), which is derived to model the distribution over the set of binary random variables, to model the edge weight at the leaf level.

The performance metrics of the proposed HiGen models are reported in Table 1, with generated graph samples presented in Figure 2. The results demonstrate that HiGen effectively captures graph statistics and achieves state-of-the-art on all the benchmarks graphs across various generation metrics. This improvement in both local and global properties of the generated graphs highlights the effectiveness of the hierarchical graph generation approach, which models communities and cross-community interactions separately. The visual comparisons of graph samples generated by the HiGen models, as well as the experimental evaluation of different node ordering and partitioning functions, are presented in Appendix D.2.

## 6 Discussion

The GRAN model can generate graphs one block of nodes at a time in an autoregressive fashion where the block size is fixed and nodes are assigned to blocks based on an ordering. However, the model's performance deteriorates as the block size increases, since adjacent nodes in an ordering may not be relevant and may belong to different clusters. Additionally, intra-block connections are not modeled separately. In contrast, our proposed method generates blocks of nodes within each community that have strong relationships and then predicts the cross-links between communities using a separate model. As a result, this approach enables the model to capture both local relationships between nodes within a community and global relationships across communities, resulting in improved expressiveness of the graph generative model.

The proposed hierarchical model allows for highly parallelizable training and generation. Specifically, let $n_c$ be the size of the largest graph cluster, then, it only requires $\mathcal{O}(n_c \log n)$ sequential steps to generate a graph of size $n$.

**Node ordering sensitivity** The predefined ordering of dimensions can be crucial for training autoregressive (AR) models Vinyals et al. (2015), and this sensitivity to node orderings is particularly pronounced in autoregressive graph generative model Liao et al. (2019); Chen et al. (2021). However, in the proposed approach, the graph generation process is divided into the generation of multiple small partitions, performed sequentially across the levels, rather than generating the entire graph by a single AR model. Therefore, given an ordering for the parent level, the graph generation depends only on the permutation of the nodes within the graph communities rather than the node ordering of the entire graph. In other words, the proposed method is invariant to a large portion of possible node permutations, and therefore the set of distinctive adjacency matrices is much smaller in HiGen. For example, the node ordering $\pi_1 = [v_1, v_2, v_3, v_4]$ with clusters $\mathcal{V}_{\mathcal{G}_1} = \{v_1, v_2\}$ and $\mathcal{V}_{\mathcal{G}_2} = \{v_3, v_4\}$ has a similar hierarchical graph as $\pi_2 = [v_1, v_3, v_2, v_4]$, since the node ordering within the communities is preserved at all levels. Formally, let $\{\mathcal{C}_i^l \ \ \forall i \in \mathcal{V}_{\mathcal{G}^{l-1}}\}$ be the set of communities at level $l$ produced by a deterministic partitioning function, where $n_i^l = |\mathcal{V}(\mathcal{C}_i^l)|$ denotes the size of each partition. The upper bound on the number of distinct node orderings in an HG generated by the proposed process is then reduced to $\prod_{l=1}^{L} \prod_i n_i^l!$. [3]

## 7 Conclusion

The proposed HiGen framework generates graphs in a hierarchical and block-wise manner, leveraging the inherent hierarchical structure present in real-world graphs. By decomposing the generation process into separate and parallel generation of communities and bipartite sub-graphs, it combines the benefits of one-shot and AR graph generative models. Experimental results on benchmark datasets demonstrate that HiGen achieves state-of-the-art performance across various generation metrics. The hierarchical and block-wise generation strategy of HiGen enables scaling up graph generative models to large and complex graphs, opening up opportunities to extend it to newer generative paradigms such as diffusion models.

---

[3]It is worth noting that all node permutations do not result in distinctive adjacency matrices due to the automorphism property of graphs Liao et al. (2019); Chen et al. (2021). Therefore, the number of node permutations provides an upper bound rather than an exact count.

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
