# OpenReview forum: "HiGen: Hierarchical Graph Generative Networks"
_NeurIPS.cc/2023/Conference — Submitted to NeurIPS 2023_

### Official Review · Reviewer_zioq · 2023-06-08

**Soundness:** 3 good
**Presentation:** 3 good
**Contribution:** 2 fair
**Rating:** 6
**Confidence:** 4

**Summary:**

This work proposes another auto-regressive-based graph generative model similar to GRAN. The authors propose a hierarchical generation scheme to un-coarse a graph level by level. In each non-leaf level, the abstract graph is weighted both in nodes and edges. A node represents a community, and its weight represents how many edges should be inside the community. An edge is the "connection" between two communities, and its weight represents how many edges should exist between the two communities. The weights of each community are generated through a stick-breaking process. And the number of communities is automatically decided by it. The structure within the community is generated using an AR model. Then the edges between communities are generated using GNN.

**Strengths:**

1. The assumption makes sense, and the model decomposition is quite convincing.
2. This method can indeed improve generation efficiency by only auto-regressively generating the diagonal blocks of the adjacency matrix and using GNN (which has O(M) runtime) to predict the off-block entries.
3. The method is simple and straightforward.

**Weaknesses:**

see questions below

**Questions:**

1. The partition function is heuristics (see experiment) and I think this is very important for training a good model, the author should elaborate more on how to generalize it to the case where L>2.
2. Optimization detail is missing, I'd like to see how the model is being optimized and what the data format is for every level. Do you need to train a model for every level?
3. It would be nice to see the runtime analysis since it's one of the motivations mentioned in the introduction.
4. The experiment only shows model performance for the case where L=2, additional experiments should be included. Otherwise, it's just a small modification from GRAN and will limit the contribution. Also, I suggest the author experiment on even larger graphs and compare the model performance to [1,2,3]
5. Since the edges of the inter-community are generated in parallel, I am concerned about the expressivity of the model due to the edge independence [2]. It would be nice to see some discussion on it.

Some related works are missing:

[1] Rendsburg, Luca, Holger Heidrich, and Ulrike Von Luxburg. "Netgan without gan: From random walks to low-rank approximations." International Conference on Machine Learning. PMLR, 2020.

[2] Chanpuriya, Sudhanshu, et al. "On the power of edge independent graph models." Advances in Neural Information Processing Systems 34 (2021): 24418-24429.

[3] Haefeli, Kilian Konstantin, et al. "Diffusion Models for Graphs Benefit From Discrete State Spaces." arXiv preprint arXiv:2210.01549 (2022).

[4] Chen, Xiaohui, et al. "Efficient and Degree-Guided Graph Generation via Discrete Diffusion Modeling." arXiv preprint arXiv:2305.04111 (2023).

[5] Kong, Lingkai, et al. "Autoregressive Diffusion Model for Graph Generation." (2023).

**Limitations:**

1. one possible limitation is the edge independency of the model.
2. The model has made a strong assumption that the graph should have a community structure, while the experiment datasets are relatively small and may not have such a structure.

---

> ### Author Rebuttal · Authors · 2023-08-09
>
>
>
> **Q1, Q4)**
> The Louvain algorithm, used as a partitioning function, is able to provide coarsened graphs at multiple levels of abstractions and we spliced out the intermediate levels to achieve HGs of size $L=2$. For the datasets reported in Table 1 of the paper, two levels of abstraction was enough as it provided clusters of reasonable sizes (refer to cluster size analysis in the "author rebuttal" section). We also conducted experimental studies for *3D point cloud* dataset with graphs up to 5K nodes and with $L=3$ levels. The results outlined in the "author rebuttal" section effectively highlight the model's performance in managing deeper hierarchical models for large graphs.
>
>
> **Q2)**
> We optimize the GNN models for all levels jointly; however, it's worth noting that these models are not shared across communities, inter-community components, parent graphs, or levels. As a result, optimization and training can be conducted independently and in parallel for each level.  In our joint implementation, the training can also be performed in parallel across levels but generation needs to be performed sequentially.
>
> For the training, we first sample a batch of $b$ HGs and then for each of these samples, we randomly sample $s$ subgraph of the communities at each level. Therefore, together with the single augmented graph for the inter-communities, defined in line 202, we will have data format $\\{ \hat{\mathcal{G}}^{l}, \hat{\mathcal{C}}^{l}\_{1}, …, \hat{\mathcal{C}}^{l}\_{s} \\}$ for level $l$ of a HG. Since we randomly sampled the subgraphs we estimate the conditional generative probability for community
> $\\mathcal{C}^{l}\_{i} $ by averaging the loss function over all the subgraphs in that community multiplied by the size of the cluster, therefore
> $$ p(\\mathcal{C}\_\{i\}^l  | \\mathcal{G}^{l-1}) \\approxeq n\_{\\mathcal{C}\_{i}} *
> mean ( [ p(u\_t (\\hat{\\mathcal{C}}^{l}\_{j}), \~ \\forall \~ \\hat{\\mathcal{C}}^{l}\_{j} \\in {\\mathcal{C}}^{l}\_{i} ] )$$
> , where $p(u\_t )$ is defined in eq. (6).
> The loss function for cross-community is straightforward as all of them are included.
>
> **Q3)**
> The average sizes of the largest cluster and some other statistics of the datasets are reported in the "author rebuttal" section.
>
> **Q5)**
> In comparison to independent models, ours employs multinomial and mixture models instead of just Bernoulli models. Consequently, the edge probabilities of inter-community components aren't treated as entirely independent. The performance of our model on the tested datasets indicates that this potential issue didn't significantly impact the results. This can also be explained by the fact that, based on [2], the edge independence is more crucial in denser community generations, whereas it's less significant in sparser inter-communities in our hierarchical approach.
>
> To address this challenge, an alternative solution is to generate inter-community components sequentially in an autoregressive (AR) manner. We implemented this approach for the 3D point cloud dataset, as detailed in Appendix D.1. However, this comes at the expense of sacrificing parallelism and some of the acceleration provided by our proposed model. Nevertheless, even with these trade-offs, this approach remains considerably faster and more efficient than GRAN.

---

> > ### Comment · Reviewer_zioq · 2023-08-18
> >
> > My concerns are mostly addressed by the responses. And I am willing to increase my rating.

---

### Official Review · Reviewer_6GcD · 2023-06-26

**Soundness:** 3 good
**Presentation:** 2 fair
**Contribution:** 3 good
**Rating:** 4
**Confidence:** 4

**Summary:**

The paper proposes HiGen a hierarchical generative graph model. The model consists of a clustering process (Louvain), followed by a GNN model (GraphGPS) to estimate probabilities. The generative process is separated by communities and bipartite sub-graphs.

**Strengths:**

The paper seems original. The proposition of a new model is always an important contribution. Even though the new model is the combination of a clustering process and GNN. The combination of both ideas is interesting.

The theoretical quality of the demonstrations is good. Most of them seem fine and no errors were observed during the revision.

Parts of the papers are quite clear. Figure 1 really helps to understand the main idea of the paper. However, there is room for improvement.

The significance of the paper is high, it seems that this new model is able to reproduce the mean of the distribution quite correctly in comparison to other state-of-the-art methods, as it is shown in the results.

**Weaknesses:**

The state of the art can be improved. The paper mentions "there exists no data-driven generative models specifically designed for generic graphs that can effectively incorporate hierarchical structure.". Neville et al. focused on this type of work, generating several papers related to hierarchical graph models (doi.org/10.1145/3161885, doi.org/10.1145/2939672.2939808, doi.org/10.1007/s10618-018-0566-x).

Parts of the paper are closed related to mKPGM (doi.org/10.1145/3161885). In both cases there is a hierarchical structure, both have the idea of a super-node at the higher level, and the sampling process is also based on a multinomial distribution (doi.org/10.1007/s10618-018-0566-x). Please take a look at the sampling process proposed, because it has similarities to the proposition of this paper, and the authors claimed to sample a network with billions of edges in less than two minutes.

The paper must state its main contribution. In the beginning, it seems to be the model, but after reading the paper, it seems to be the sampling process. Unfortunately, both of them have different issues.

If the main contribution is the model, then the paper should improve the modeling of the main network and be fairly compared in the experiment section against other baselines (not just the mean of the distribution). The main models consider $\ell$ hierarchies, but just two are applied. It is also not clear how the final probabilities are obtained.

If the main contribution is the sampling process, there are some issues too. The time complexity of the generative model claims to be O(n_c \log n), but this is not demonstrated. The results of the paper are focused on the modeling of networks, not the sampling process. For example, there are no empirical results about the time complexity, and the largest networks have some thousand nodes, rather than millions.

I understand that the papers follow the experimental setup and evaluation metrics of Liao et al. However, this methodology must be stated in the main paper, otherwise, the experiments of the main paper are not reproducible.

The results of Table 1 are difficult to read because of the lack of explanations. There are no details on the separation of the data in the main paper. I understand that this is explained in the supplementary material (80% for training and 20% for testing), but it must be considered in the main paper too. Moreover, I do not know if the values are the average over the 20% of the testing graphs or, if you just considered it as a single distribution. In the first case, please add the standard deviation, to see if the difference at statistically significant.

Section 5 claims: "The results demonstrate that HiGen effectively captures graph statistics". Considering that, generally speaking, MMD estimates the distance between the means of two distributions, I suggest you change it to "The results demonstrate that HiGen effectively captures the mean of the graph statistics". Given the use of MMD, you can not determine if the other part of the distribution are correctly estimated.

The conclusions state that HiGen "enables scaling up graph generative models to large and complex graphs" but this is not demonstrated.

**Questions:**

-What are the similarities and differences between this sampling process and the sampling process proposed at doi.org/10.1007/s10618-018-0566-x? Can you use that sampling process to speed up your generative process?
-Why did you consider such small networks?
-Can you, at least, empirically demonstrated the time complexity of your model?
-how did you compare the final distribution for the MMD? Did you consider the average of each network, or do you make a single distribution considering all networks?


**Limitations:**

No, the authors did not consider the limitations of the proposed model. For suggestions, please check weaknesses.

---

> ### Author Rebuttal · Authors · 2023-08-10
>
> **mKPGM papers:**
> Compared to our proposed model, Kronecker product graph models lack the depth and complexity of deep neural networks and graph neural networks. Consequently, their capacity for modeling complex relationships in graphs is notably restricted. These models mainly concentrate on capturing specific statistical characteristics of graphs, such as degree distribution, rather than the complex structures present in the data.
>
> In contrast, \HiGeN adeptly models both intra-links and cross-links within graphs by employing separate GNN models. Notably, these models consider the parent graph's structure when determining edge distributions. This enables \HiGeN to capture the hierarchical nature of graphs and the interactions between different levels of the hierarchy. This approach offers a more expressive and flexible framework for graph generation compared to Kronecker product models.
>
> **main contribution:**
> Our proposed approach focuses on generating clusters of interconnected nodes within each community, capturing local relationships effectively. Additionally, it predicts cross-links connecting different communities using a separate model. This strategy allows our model to simultaneously capture both fine-grained local connections and broader global relationships, thereby leveraging the inherent hierarchical structure prevalent in real-world graphs.
>
> Furthermore, we extended our experimental analysis to the 3D point cloud dataset, comprising graphs of up to 5K nodes and employing a hierarchical level ($L=3$). The outcomes, detailed in the "author rebuttal" section, emphasize the model's capability to handle deeper hierarchies for larger graphs.
>
> *“It is also not clear how the final probabilities are obtained.” :* In this work the probability of the edges of the graph are modeled in hierarchical fashion and for community graphs and inter-community components according to equations (6) and (8), respectively.  In contrast to mKPGM models, we used deep GNNs to model the parameters of these high dimensional probabilities. Besides offering a novel graph generative model, the hierarchical and modular structure offers parallel sampling of the distribution.
>
>
> **MMD metrics:**
> There seems to be a misunderstanding regarding MMD.
> The maximum mean discrepancy (MMD) is a distance-measure (or discrepancy measure) between two distributions using their sets of samples of them. It provides an efficient distance metric between two distributions by using first Wasserstein distance (earth mover’s distance) as the kernel hence it preserves all of the statistical features of arbitrary distributions, not just the mean of the distribution. You can find a detailed explanation of the MMD for graph statistics in Section 4.3 of [1].
>
>
> **Experimental Results:**
> In this work, we followed the evaluation metrics of SOTA models such as GRAN, SPECTRE, DIGRESS and GDSS by reporting the MMD that measure the distance (discrepancy) between the distributions of the generated and test graph statistics, such as degree, i.e.  $p\_{gen}(degree)$ and $p\_{test}(degree)$. So in all of these works, one distribution is assumed for each statistic and the MMD is computed based on the samples of this distribution.
>
> The details of the experimental setup and the distinction between training and test sets will be included in the final version of the main paper.
>
> Scalability: While recent models like SPECTRE, DIGRESS, and GDSS have demonstrated efficiency for graphs comprising several hundred nodes, our work showcases the scalability of the proposed models to accommodate graphs with several thousand nodes. It's important to note that this scalability isn't limited to the specified range and the proposed model has potential for larger graphs.
>
>
> [1] Jiaxuan You et all. Graphrnn: Generating realistic graphs with deep auto-regressive models. In ICML, pp. 5694–5703, 2018.

---

> > ### Comment · Reviewer_6GcD · 2023-08-17
> > **Author rebuttal**
> >
> > Thanks for the response, I have read this response and others carefully. I know the way that is evaluated. I am just saying than the paper must try to be complete by itself.

---

> > > ### Author Response · Authors · 2023-08-21
> > >
> > > Thank you for your insightful feedback. During the initial submission, due to the the page limit constraints, we focused on incorporating the main discussions and following the experimentation approach of the baseline paper in Graph Generation, in order to convey the core concepts and contributions of our work. Since we have one extra page in the final version, we will integrate the supplementary analysis, comprehensive clarifications and new results that have been thoroughly discussed in the rebuttal period.

---

### Official Review · Reviewer_FuoC · 2023-07-06

**Soundness:** 2 fair
**Presentation:** 3 good
**Contribution:** 2 fair
**Rating:** 5
**Confidence:** 4

**Summary:**

The paper introduces an innovative hierarchical method for graph generation, which employs multiple levels of graph coarsening. This approach begins with the first level, representing the most coarse graph, and progressively expands nodes and edges to form new communities and connections between the newly created nodes. At each level, nodes serve as communities for the subsequent level, and the edge weights, including both inter-community edges and self-loops, dictate the total number of edges within each community in the final graph. Consideration of independence among the generation processes of inter and intra-community edges, conditioned on the graph and edge weights from previous levels enables parallel execution of the steps, resulting in acceleration of the generation process.

**Strengths:**

1. The paper effectively utilizes hierarchical clustering to enhance the graph generation process, capitalizing on the benefits of this technique.
2. By introducing parallelization in generating distinct clusters at each level, the paper successfully minimizes the number of sequential steps required.
3. The experimental results presented in the paper demonstrate improvements across multiple datasets.
4. Paper for the most part is well-written and easy to follow.

**Weaknesses:**

1. In lines 35-36 paper mentions that this work is the first hierarchical method for generic graphs. I believe [1] is also a hierarchal method for graph generation. I understand that methods are significantly different but still it would be more accurate to highlight the unique aspects of the proposed method and consider including a comparison between the two approaches.
2. The time complexity analysis provided in the paper focuses solely on the sequential steps, neglecting to consider the computational requirements. It would be valuable to compare the overall computational workload, particularly since the proposed method utilizes the GraphGPS approach, which has a time complexity of $O(n^2)$, in contrast to conventional GNN methods with a complexity of $O(n+m)$. Including such a comparison would provide a more comprehensive analysis.
3. The paper lacks a study examining the distribution of community sizes during the generation process across different datasets. Addressing this limitation by investigating and reporting the distribution of community sizes would enhance the understanding of the method's behavior and its adaptability to various datasets.
4. The paper uses a more advanced GNN compared to methods like GRAN, raising the question of how much of the observed progress is solely due to the change in the GNN architecture. Conducting an ablation study specifically focused on the GNN architecture used would provide valuable insights into its individual contribution to the overall performance of the method.
5. The evaluation metrics commonly employed for graph generative models have their limitations, as discussed in [1] and [2]. It is important to consider these limitations in the evaluation process. The paper mentions the use of random GNNs as an alternative evaluation method, but this approach is only used in the appendix for a few experiments. I would suggest using this in the main body and comparing all models using this metric. [Additionally/Optionally, there are two more recent approaches, one based on contrastive training and another one based on Ricci curvatures that could be incorporated for evaluation purposes.]

[1] Shirzad, H., Hajimirsadeghi, H., Abdi, A. H., & Mori, G. (2022, May). TD-gen: Graph generation using tree decomposition. In International Conference on Artificial Intelligence and Statistics (pp. 5518-5537). PMLR.

[2] O'Bray, Leslie, et al. "Evaluation metrics for graph generative models: Problems, pitfalls, and practical solutions." arXiv preprint arXiv:2106.01098 (2021).

**Questions:**

1. From line 253 it appears that the number of levels is fixed to 2 for all datasets, is this correct? What is the average size of the largest cluster for different datasets used? This information is required for understanding the levels of improvement the model makes.
2. The formulation of Theorem 3.1 is confusing, particularly in relation to the assumption of Multinomial distributions. It seems that the theorem does not fully consider the fact that some nodes are shared during the process of distributing edges among nodes for both intra and inter-cluster components. To clarify, let's consider the scenario of learning the distribution over d-regular graphs. In the final level, the number of edges associated with a node within its cluster and the sum of edges from intra-cluster connections must be fixed. Therefore, we cannot treat them as independent, even if we know the sum of the number of edges inside a cluster and the number of edges between each two clusters. It would be beneficial to address this concern and provide further clarification on how the theorem accounts for shared nodes during edge distribution.
3. An analysis of the number of parameters in the model would be valuable information to include in the paper. Given the relatively small number of samples in the datasets used and the utilization of large networks, particularly with each hierarchy level having its own separate GNN comprising 8 layers, the complexity and capacity of the model should be carefully considered. Providing details on the number of parameters, as well as discussing potential implications of model size and dataset size, would enhance the understanding and interpretation of the experimental results.

**Limitations:**

Considering the assumed independence among the clusters and the cross-edges connecting them, it is evident that there exist certain graph distributions which the model may struggle to learn.

---

> ### Author Rebuttal · Authors · 2023-08-09
>
> **W1)**
> Compared to the proposed method, TD-gen is limited to a single level of abstraction with tree structure and its graph generation requires $O(nk)$ steps where k is the width of the tree decomposition therefore its scalability is limited to medium size graphs. This comparison will be included in the final version.
>
>
> **W2)**
> As clarified in appendix B, the quadratic complexity of GraphGPS can be mitigated by employing linear Transformers like Performer or newer models such as Exphormer, effectively reducing GraphGPS complexity to $O(n+m)$ while maintaining HiGeN's scalability. For the large 3D Point Cloud graphs, we utilized Performer and the results are detailed in the appendix, along with an attached PDF for $L=3$.
>
> Furthermore, since Transformers are parallelizable, they can take advantage of GPU acceleration and achieve high speeds when fitting within GPU memory.
>
> **W3 and Q1)**
> The average sizes of the largest cluster are reported in the general "author rebuttal". In the paper, we also compared the MMD of the clustering coefficient, which measures the extent to which nodes in a graph tend to cluster together.
>
> **W4)**
> In the experimental section, we employed the GAT model used by GRAN to derive node features for the augmented sub-graphs  $\mathbf{h}\_{\hat{\mathcal{C}}^{l}\_{i,t} } $. This explanation is missed to be added in the Appendix B.
>
> Furthermore, we customized the GraphGPS for application on graphs with augmented bipartite graphs $\mathcal{G}^{l-1} $, incorporating distinct initial edge features to differentiate augmented (candidate) edges from actual edges. GraphGPS was also utilized to acquire node features for the parent graphs.
>
> Additionally, for the Enzyme dataset, we conducted experiments where we replaced GAT with GraphGPS as the GNN model for communities. Although the results were quite close to those using GAT, these experiments were not included in the analysis, as the primary contribution and the focus of this model lies in developing a hierarchical generative framework.
>
> **W5)** In this work, we followed the evaluation metrics of SOTA models such as GRAN, SPECTRE, DIGRESS and GDSS and used the structure based metrics reported by them to compare the Higen’s results with theirs. Furthermore, an additional study comparing the performance of HiGeN on the Ego dataset using GNN-based metrics is presented in the general "author rebuttal" section.
>
> **Q1)**
> Yes, for these datasets, the number of the levels was set to 2. However, the proposed model , in conjunction with the Louvain algorithm used as a partitioning function, offers the potential for the dataset's expansion to larger graphs with greater values of $L$. The experimental outcomes for the *3D point cloud* dataset with $L=3$ are outlined in "author rebuttal" section. These outcomes effectively highlight the model's performance in managing deeper hierarchical graphs.
>
> Additionally, the average sizes of the largest cluster and other pertinent graph statistics are presented in the "author rebuttal" section.
>
> **Q2)**
> Thanks for bringing up this concern. To clarify, our model is based on the assumption that the communities are mutually independent given the parent level. Following the generation of cluster graphs, it is also assumed that the generation of each bipartite (inter-cluster component) can be modeled independent of the rest of the BPs which enables us to accelerate the final graph generation. Hence, the inter-community generation step does not require independence among the clusters and the cross-edges. Therefore, to correct this mis-intrerpretation we need to adjust equation (2) as:
> $$     p(\mathcal{G}^l | \mathcal{G}^{l-1}) \approxeq \prod p(\mathcal{C}\_{i}^l  | \mathcal{G}^{l-1}) \times \prod  p(\mathcal{B}\_{ij}^l  | \mathcal{G}^{l-1}, \\{ \mathcal{C}\_{i} \forall \mathcal{C}\_{i} \in \mathcal{G}^{l} \\})$$
>
> Therefore, the model is not based on independence of the communities and inter-cluster components. In fact, Higen captures this dependency by formulating the parameters of multinomial distribution for inter-cluster components in theorem 3.1 as a function of the already generated communities and the parent graph,
> $\mathbf{\theta}^l\_{ij} = f (\mathcal{G}^{l-1}, \\{ \mathcal{C}\_{i} \forall \mathcal{C}\_{i} \in \mathcal{G}^{l} \\})$.
>
> Given that link predictions of the inter-cluster occur after the generation of all clusters and rely on them, as demonstrated in figures 1.d and 1.e,  an expressive deep NN in Higen should be able to learn patterns such as d-regular.
>
> **Q3)**
> The number of parameters is provided in the "author rebuttal" section, illustrating that the proposed model achieves superior performance with fewer parameters compared to GRAN. This emphasizes the efficiency of hierarchically modeling communities and cross-community interactions as distinct entities. This can be explained by the fact that the proposed model needs to learn smaller community sizes compared to GRAN, which, consequently, enables us to use smaller models which make the model training faster.

---

> > ### Comment · Reviewer_FuoC · 2023-08-15
> >
> > Thank you for your dedication in responding! Some of my concerns have been addressed, but some critical ones still remain:
> >
> > 1. Regarding TD-Gen, I agree with your argument about TD-Gen's limited abstraction level. However, the time complexity part has been confusing for me. Isn't it the case that your model in the worst case has to create the whole adjacency matrix and so it can be $O(n^2)$ which is worse than $O(nk)$? (Always $k<n$)
> >
> >
> > 2. Regarding the evaluation metrics, I don't think that following the previous work's method is a must. Shortcomings of older methods have been evident in several works including [1, 2] and TD-Gen. Now that we know these issues, it's important for recent graph generative papers to consider them and address them in their work. An argument discussed in TD-Gen is the ability of the models in memorizing the train dataset. Based on Table 1 in your rebuttal pdf, considering 4 bytes/parameter we can see that in many cases models are larger than the dataset itself. This is not inherently wrong, but an analysis of overfitting should be done in this case. For an extreme case scenario assume we have a model that just has stored train data; then for sampling every time it just selects one of the train samples and gives it as output. This model will give you near zero results on MMD distances, as train and test datasets are coming from the same distribution; and also time and space complexity for this model will be even almost perfect. However, I hope that we agree that this is not a useful model and it has not been the purpose of learning generative models. As a result, we need to make sure that the model has not just memorized the train data. Likelihood could help with such analysis but due to the huge number of permutations likelihood is intractable in many cases. Using more advanced comparisons such as using Precision and Recall analysis may be helpful here. Also, a method from [3] can be used to combine more recent GNN-based evaluation metrics with older ones.
> >
> >
> > 3. Thanks for clarifying the independence assumptions. It makes it much more clear now. I still think there are cases of d-regular graphs that HiGen can't handle though. Assume we make d-regular graphs like this: We make $k$ cliques of size $d$ and then we connect each node to exactly one node outside of its clique. Now, for a large enough $d$ a natural algorithm for making denser clusters will most probably put each clique into one cluster. Clusters will be efficiently generated as all nodes inside a cluster are connected to each other. But, for the inter-cluster edges, from the coarsened graph, we can only understand how many edges are between two clusters. But, we can't possibly understand which nodes are going to connect to each other. All nodes are symmetric at this point after cluster generation. Thus, if we generate intercluster parts in parallel, a node from a cluster can not possibly understand if it has connected to a node from other BPs or not, and so uniformity can't be handled here. To be clear, I don't think that this is a huge problem about the work, many works have their limitations; however, I think the limitations should be clearly stated in the paper.
> >
> > I appreciate the authors' detailed responses and the new results they have provided, however, since some of my key concerns are remaining, I will keep my score.
> >
> > [1] O'Bray, Leslie, et al. "Evaluation metrics for graph generative models: Problems, pitfalls, and practical solutions." arXiv preprint arXiv:2106.01098 (2021).
> >
> > [2] Thompson, Rylee, et al. "On evaluation metrics for graph generative models." arXiv preprint arXiv:2201.09871 (2022).
> >
> > [3] Shirzad, H., Hassani, K., & Sutherland, D. J. (2022). Evaluating graph generative models with contrastively learned features. Advances in Neural Information Processing Systems, 35, 7783-7795.

---

> > > ### Author Response · Authors · 2023-08-17
> > > **GNN-based metrics comparison**
> > >
> > > Thank you for your feedback and for considering our response. In the following I addressed your concerns:
> > >
> > > 1 ) In the worst-case scenario, if no partitioning occurs, our model operates as a hierarchical graph with a single cluster (equivalent to L=1). As the model generates one node at a time, it essentially reduces to a GRAN-like process with $O(N)$ generation steps. Thus, the worst-case generation step is $O(N)$.
> > >
> > > Furthermore, it's worth noting that a key distinction from TD-gen is our model's ability to employ different generation models for clusters and inter-clusters, adding to its versatility and performance capabilities."
> > >
> > > 2 ) As the proposed framework extends GRAN and uses that as a building block for community generation, we conducted experiments comparing the performance of both models across a range of structure-based and GNN-based metrics. The results are presented in the table below:
> > >
> > > | Model         	| Deg. $\downarrow$ | Clus. $\downarrow$ | Orbit $\downarrow$ | Spec., $\downarrow$ | GNN MMD $\downarrow$ | GNN F1 PR $\uparrow$ | GNN F1 DC $\uparrow$ |
> > > |-------------------|:-----------------:|:------------------:|:-----------------:|:-------------------:|:-------------------:|:-------------------:|:-------------------:|
> > > | *Enzyme*       	|               	|                	|               	|                 	|                 	|                 	|                 	|
> > > | **GRAN**      | 8.45e-03       	| 2.62e-02       	| 2.11e-02      	| 3.46e-02         	| 0.0663          	| 0.950           	| 0.832           	|
> > > | **HiGen-m**    | 6.61e-03       	| 2.65e-02       	| 2.15e-03      	| 8.75e-03         	| 0.0215          	| 0.970           	| 0.897           	|
> > > | **HiGen**      | 2.31e-03       	| 2.08e-02       	| 1.51e-03      	| 9.56e-03         	| 0.0180          	| 0.978           	| 0.983           	|
> > > | *Stochastic block model* |          	|                	|               	|                 	|                 	|                 	|                 	|
> > > | **GRAN**      | 0.0159         	| 0.0518         	| 0.0462        	| 0.0104           	| 0.0653          	| 0.977           	| 0.86            	|
> > > | **HiGen-m**    | 0.0017         	| 0.0503         	| 0.0604        	| 0.0068           	| 0.154           	| 0.912           	| 0.83            	|
> > > | **HiGen**      | 0.0019         	| 0.0498         	| 0.0352        	| 0.0046           	| 0.0432          	| 0.986           	| 1.07            	|
> > > | *Ego*          	|               	|                	|               	|                 	|                 	|                 	|                 	|
> > > | **GraphRNN**      | 9.55e-3        	| 0.094          	| 0.048         	| 0.025            	| 0.0972          	| 0.86            	| 0.45            	|
> > > | **GRAN**      | 7.65e-3        	| 0.066          	| 0.043         	| 0.026            	| 0.0700          	| 0.76            	| 0.50            	|
> > > | **HiGen-m**    | 0.011          	| 0.063          	| 0.021         	| 0.013            	| 0.0420          	| 0.87            	| 0.68            	|
> > > | **HiGen**      | 1.9e-3         	| 0.049          	| 0.029         	| 0.004            	| 0.0520          	| 0.88            	| 0.69            	|
> > >
> > > The table includes the average of random-GNN-based metrics [1] over 10 random Graph Isomorphism Network (GIN) initializations, including metrics such as MMD with RBF kernel (GNN MMD), the harmonic mean of improved precision+recall (GNN F1 PR), and harmonic mean of density+coverage (GNN F1 PR). Here, we reported the TV distance for the structure-based statistics. Moreover, in the following table (table 3 of the attached pdf + Frechet Distance (FD)) the GNN-based performance metrics (GNN MMD and FD) of HiGen are compared against the baselines reported in [3] for Ego dataset where the Gaussian EMD kernel was used for structure-based statistics.
> > >
> > > | Model    |  Deg. | Clus.  | Orbit | GNN MMD | FD $\downarrow$ |
> > > |----------|---------------------|--------------------|------------------|--------------------|----------------|
> > > | **GraphRNN**  | 0.0768 | 1.1456 | 0.1087 | 0.6827 | 90.57 |
> > > | **GRAN**      | 0.5778 | 0.3360 | 0.0406 | 0.2633 | 489.96 |
> > > | **GDSS**      | 0.8189 | 0.6032 | 0.3315 | 0.4331 | 60.61  |
> > > | **DiscDDPM**  | 0.4613 | 0.1681 | 0.0633 | 0.1561 | 42.80  |
> > > | **DiGress**   | 0.0708 | 0.0092 | 0.1205 | 0.0489 | 18.68  |
> > > | **EDGE**      | 0.0579 | 0.1773 | 0.0519 | 0.0658 | 15.76  |
> > > | **HiGen**    | 0.0472 | 0.0031 | 0.0387 | 0.0454 | 5.24   |
> > >
> > > Regarding your concern about model sizes, we have comprehensively addressed this concern in the official comment titled *On Model sizes*. Despite the notably smaller or equal model sizes, HiGen consistently surpasses GRAN's performance across various metrics.
> > >
> > > [1] Thompson, Rylee, et al. "On evaluation metrics for graph generative models." arXiv preprint arXiv:2201.09871 (2022).
> > > [2] Chen, Xiaohui, et al. "Efficient and Degree-Guided Graph Generation via Discrete Diffusion Modeling." arXiv preprint arXiv:2305.04111 (2023).

---

> > > > ### Author Response · Authors · 2023-08-17
> > > > **Edge independent in generation of inter-cluster components**
> > > >
> > > > 3 ) Providing a brief contextual background for readers to better follow the discussion, the initial question was about the independence of communities and inter-cluster components, a concern that has been addressed in the response. The subsequent question pertains to the parallel and independent generation of inter-cluster components (bipartites).
> > > >
> > > > We have addressed a relevant concern of edge independence generation of inter-cluster components in response to another reviewer (https://openreview.net/forum?id=SlXKgBPMPn&noteId=1BIfj6f3qU). In addressing your concern and the scenario you presented, it is plausible to design a set of adversarial examples for any learning algorithm for which it exhibits suboptimal performance. However, the critical question is how likely  and realistic are such scenarios, such as the symmetrical and d-regular instances, in the real-world context. Although it could potentially be a limitation of the proposed graph generative model, it's worth noting that the significance of edge independence is more pronounced in high densities graphs such as community generations [3], whereas it's less significant in sparser inter-communities in our hierarchical approach. This is evident by the performance improvement of HiGen observed in the experiments.
> > > >
> > > > To address the dependencies among bipartites, viable modifications can be made to the framework. An alternative approach involves generating inter-community components sequentially in an autoregressive (AR) fashion. This approach was implemented for the 3D point cloud dataset, as elaborated in Appendix D.1. and its variant with L=3. It's important to note that even in such an approach, bipartites not adjacent to each other can still be generated in parallel, instead of being entirely sequential. The adjacency of two bipartites can be determined based on the edge connectivity of the parent level and whether their parent edges are connected to the same node. Another prospective solution involves training a Provably Powerful Graph Network (PPGN) as a refinement step post bipartite generation. This network has an expressive power that aligns with that of a 3-WL test, a strategy employed in SPECTRE [4].
> > > >
> > > > In situations where the issue of edge independence within a bipartite is a concern, it's important to recognize that both GRAN and the bipartite generation of HiGen add a cut-set of edges at a time. However, in the scenario of d-regularity, when GRAN introduces a new node into a clique, it assigns equal probabilities to all possible edges and may connect all links due to symmetry. On the other hand, HiGen possesses information regarding the number of edges within that cut-set and employs a multinomial probability distribution to model the joint distribution.
> > > >
> > > > In the final version, we will include a comprehensive discussion concerning such limitations and potential solutions.
> > > >
> > > > We hope this clarification addresses your concerns. We highly value your insights and suggestions and are committed to addressing any remaining concern you may have.
> > > >
> > > > [3] Chanpuriya, Sudhanshu, et al. "On the power of edge independent graph models." Advances in Neural Information Processing Systems 34 (2021): 24418-24429.
> > > > [4] Martinkus, Karolis, et al. "Spectre: Spectral conditioning helps to overcome the expressivity limits of one-shot graph generators." International Conference on Machine Learning. PMLR, 2022.

---

> > > > > ### Comment · Reviewer_FuoC · 2023-08-20
> > > > >
> > > > > Thanks for your thorough responses and new experimental results!
> > > > >
> > > > > 1) Regarding the issue of memorizing the training dataset (overfitting), Table 2 in TD-Gen paper [1] shows that GRAN appears to be more susceptible to overfitting, so having it as the only baseline on this part will not be sufficient to prove the point. It would be beneficial to include a comparison with other baseline models. While I recognize that your model is generally smaller than GRAN, it still often surpasses the dataset's size. I suggest that the authors address this concern in their paper. Given the limited discussion time and approaching deadline, I think reporting average likelihood values, with any set of permutations you have used, on train and test datasets can be helpful here. Or alternatively comparing with GraphRNN on Ego dataset and with "GNN F1 PR" and "GNN F1 DC" metrics can be informative.
> > > > >
> > > > > 2) About the example of the d-regular graphs and the power of the method; first of all, I wanted to clarify again that I don't think this means the model is not effective at all, indeed your experiments shows its effectiveness on several datasets, still, I believe limitations should be clearly stated in the paper. The extent of the models' capabilities is a crucial aspect that should not be overlooked. My example was not a very special adversarial case, but, it was just a simple sample that could convey my concern. The main point is that the algorithm can fall short in capturing a highly localized graph property such as node degrees, having current assumptions. In real-world graphs, there exists a lot of short and long-range dependencies among edges, raising the question that how your network can model these dependencies if the clustering algorithm does not effectively cluster the dependent edges into the same cluster or bipartite. I understand that certain datasets may not necessitate this, or where clustering proves highly effective; however, my concern is mainly around the misinterpretation that can happen from reading the paper. Particularly, Theorem 3.1 seemingly implies that the independence assumption universally holds when conditioning on upper layer weights. While the decomposition of the Multinomial distribution is valid, but modeling of the new layer as distributing the edge weights among the connections in the new layer is the part that seems to be oversimplification, leading to this result. This formulation assumes edge weights as independent variables and overlooks that edges share endpoints, thus not being generally independent. This has been more clear in the rebuttal, but I would like to ask authors how they are planning to fix the misinterpretations that can raise from this theorem in their paper. Thanks for accepting having this discussion in your paper, having this discussion and clarifying the theorem will resolve my concern at this part.
> > > > >
> > > > > I am willing to raise my score uppong receiving one of the evidences mentioned on point (1) indicating model is not overfitted on the train dataset and a plan to clarify the model's power in the paper.
> > > > >
> > > > > [1] Shirzad, H., Hajimirsadeghi, H., Abdi, A. H., & Mori, G. (2022, May). TD-gen: Graph generation using tree decomposition. In International Conference on Artificial Intelligence and Statistics (pp. 5518-5537). PMLR.

---

> > > > > > ### Author Response · Authors · 2023-08-21
> > > > > >
> > > > > > Thank you for your response and constructive feedback.
> > > > > > To address (1) we have revised the table for the Ego dataset in the previous response (https://openreview.net/forum?id=SlXKgBPMPn&noteId=omwuYT8kO3) to include GNN F1 PR (harmonic mean of improved precision+recall), GNN F1 DC (harmonic mean of density+coverage) and FD (Frechet Distance) for GraphRNN. As indicated by these results, HiGen outperforms GraphRNN and GRAN in all of these metrics which also provide evidence against overfitting to the training dataset.
> > > > > >
> > > > > > To address the misinterpretation about the independence of communities and inter-cluster components, we will update equation (2), as stated in our first response, and clarify the dependency of the BP generation on already generated clusters and the parent graph together. To clarify the model's power and its potential limitation regarding edge independence among inter-cluster components, we will summarize these conversations and add discussion on the edge independence of inter-cluster components together with the potential solutions and modifications to the model, such as AR generation of inter-cluster components that was discussed in the last response. Furthermore, we will include detailed insights into the model sizes and the Sampling Speed Comparison as outlined in response: https://openreview.net/forum?id=SlXKgBPMPn&noteId=5vRIl0B01a in the final version.

---

> > > > > > > ### Comment · Reviewer_FuoC · 2023-08-21
> > > > > > >
> > > > > > > Thank you for providing new experimental results and a plan to resolve the misinterpretation issue. I am now more toward acceptance. I increased my score in the main review.

---

### Official Review · Reviewer_1z1k · 2023-07-11

**Soundness:** 2 fair
**Presentation:** 1 poor
**Contribution:** 3 good
**Rating:** 5
**Confidence:** 3

**Summary:**

The paper introduces a graph generative model that is analogously structured as the inverse process of graph pooling, where the model first split a single node into a metagraph. This metagraph is further partitioned by utilizing a multinomial scheme, which allows for the division of nodes and edges into intra-community and inter-community connections. The proposed model's performance is evaluated on several benchmarks using various metrics, demonstrating state-of-the-art results.

**Strengths:**

1. The approach of initially generating a graph's skeleton and subsequently refining its details is a novel and intuitively logical motivation for the proposed model.

2. The proposed methods have demonstrated state-of-the-art performance on some widely adopted benchmarks.

**Weaknesses:**

1. Some important technical aspects in the paper may require additional clarification or more detailed elaboration. Here are the major concerns regarding specific aspects:

    (1) Can the authors please provide more information on the loss function utilized in the model?

    (2) How is the weight on level 0 determined during model inference?

    (3) On line 188, the node embedding matrix $\mathbf{h}_{\hat{C}}$ is referenced without being defined. Could the authors please explain how this matrix is generated from the node and edge embeddings of prior levels?

    (4) Could the authors please elaborate on how the graph neural network (GNN) is utilized throughout the entire process?

2. The utilization of notations in the paper has resulted in a significant amount of confusion. There are two main issues that need to be addressed:

    (1) Inconsistent notations caused by reusing the same symbols: One notable example is the letter "t" used at lines 187-188, which has multiple interpretations. In $\hat{C}_{i,t}^l$, "t" represents the "t-th" step in the stick-breaking process. In $h{(t, s)}$, "t" denotes the node that is associated with community "i". Moreover, when referring to the node matrix size as "$t \times d_h$", it indicates the total number of nodes in community "i". These varying interpretations of the same notation can lead to confusion and should be clearly distinguished or explained consistently throughout the paper.

    (2) Notations used without being defined: An example is the "r" symbol in Figure 1 (c). Although it is assumed to represent the acronym for "remaining (edges)," its precise definition is not explicitly provided in the paper. To enhance clarity, it would be beneficial to define such notations explicitly or provide a glossary of symbols and their corresponding definitions.

3. In the paper, the specific method for determining the number of mutually exclusive events (i.e., the edges split from the same parent node) when modeling the partition weights using a multinomial distribution is not explicitly mentioned. This aspect requires further clarification or explanation. The paper should provide details on how the number of events is determined, whether it is considered a fixed parameter based on the model's architecture or if it is treated as a latent variable to be inferred during the training process.

**Questions:**

1. Please address my questions in the previous section.

2. At line 188, the paper defines the feature for an edge $(t, s)$ as the difference between node features. This implies that the features for any self-edge would be zero, as the difference between a node's feature and itself would result in a zero value. Regarding Eq(7), it states that the expected partition weight received by each edge is determined by concatenating the edge feature and the parent node feature, which remains constant for all edges split from the same parent node. Considering these points, if all self-edges have identical features, they are expected to receive the same partition weights. Could authors please confirm or repute the above analysis?

3. As a consequence of similar partition weights across the self-edges split from the same parent node, when these self-edges are further partitioned into intra-community connections, it could lead to similar sizes and connection densities across the communities. Does it align with the characteristics of real-world graph generation that the model intends to apply to? In other words, how well is the proposed model in generating graphs whose communities has varying sizes and connection densities.

4. I have some questions regarding the rationale behind introducing theorem 3.3, especially considering the prior introduction of lemma 3.2. It seems to me that theorem 3.3 primarily focuses on consolidating the two levels of split into a single function. I kindly request the authors to provide clarification regarding the significance and necessity of theorem 3.3 in the overall context of the paper.

**Limitations:**

1. Please refer to questions 2 & 3.

2. The model is built upon the assumption that the graph contains underlying communities. While this assumption can aid in generating higher-quality graphs with evident community structures, it may come at the expense of the generation quality for graphs where the community structures are less apparent. It would be intriguing to explore how the quality of generated graphs varies with changes in graph modularity or other community metrics.

---

> ### Author Rebuttal · Authors · 2023-08-08
>
> **W1.1 )** Binomial and Multinomial are from the exponential family distribution and hence their log-likelihood reduces to Bregman divergence []. The binomial likelihood is a general form of Bernoulli likelihood, and multinomial likelihood is a general form of multinoulli log likelihood. These details will be added to the appendix for clarification
>
> W1.2)
> As explained in lines 123-125, we estimate $p(w^0)$ by computing the empirical distribution (histogram) of $w\_0$ in the training set. Note that, it is the number of edges in the graph with binary weights, or $w\_0=m$,  and it is the sum of the edge weights for the graph with integer-valued weights. Given the vector $p$ is the probability of all possible values of $w\_0$, at the inference time (sample generation) we sample w0 from a multinouli distribution with $PMF=p$.
>
> W1.3 & W1.4) Node embedding $\mathbf{h}\_{\hat{\mathcal{C}}^{l}\_{i,t} }$  , also called node features in the paper, are learned by GNN models (line 189) applied on the augmented sub-graph at the same level $\hat{\mathcal{C}}^{l}\_{i,t} $ ,
> so we can write
> $\mathbf{h}\_{\hat{\mathcal{C}}^{l}\_{i,t} } = GNN^l\_{com} ( \hat{\mathcal{C}}^{l}\_{i,t} )$
> (this equation will be added in the final version for clarification).
> Note that we assumed the node features are functions of the sub-graph at level $l$, not the node and edge embedding of prior levels.
> After obtaining edge embedding and sub-graph embedding (graph level representation), they are concatenated with the node features of its parent node to enrich edge embedding and sub-graph embedding used in eq. (7) for calculating edge probabilities (line 195). This is how the final probability of the new edges are estimated based on the  $\mathbf{h}\_{\hat{\mathcal{C}}^{l}\_{i,t} }$ and its parent level.
> For inter-community (bipartite) we obtain node features as  $\mathbf{h}\_{\hat{\mathcal{B}}^{l}\_{i,j} } = GNN^l\_{bp} ( \hat{\mathcal{G}}^{l}  )$ and  $\mathbf{h}\_{\mathcal{G}^{l-1} } = GNN^l (\mathcal{G}^{l-1}   ) $,  respectively, where $\hat{\mathcal{G}}^{l}$ is defined in line 202.
>
>
> As the main focus of this section is on designing a hierarchical and auto-regressive model for community and inter-community generation, the Node Feature Encoding models are explained after them and in the appendix.
>
>
> 2.1) Since the community $i$ recursively by adding a new node at each time step $t$ to the already generated sub-graph $\hat{\mathcal{C}}^{l}\_{i,t}$ , therefore it is equal to the total number of nodes in community $i$ .
> Therefore, we denoted the new node that is added at time step $t $  by $ v\_t $.
>
> To illustrate, the 4th nodes in figure 4.d for community generation are added at time step $t=4$ and their corresponding edges to the already generated communities are decided at this step. The figure in the attached pdf for auto-regressive community generation will also help resolving this confusion.
>
> 2.2)  As you mentioned, $r$ denotes the remaining weights. Indeed it is defined in the caption of Figure 1 but with a typo so the correction will be "... fraction of the remaining weights $r\_m$ is allocated to the $m$-th row ... ". This variable is equal to $r\_m = w - \sum \_{i < m} v\_m$ in equation 3.3.
>
> 3 ) The total number of edges (events) is established by the weight of corresponding edges in the parent graph, as elaborated in Section 2 and further detailed in Appendix (lines 464 - 468). This value serves as the initial weight for the remaining weights during community generation. At each step of generation, this value decreases. Therefore for community $i$ at level $l$, the remaining weights are calculated as follows:  $ r\_0 = w^{l-1}\_{00}, r\_1 = r\_0 - v\_1, r\_2 = r\_1 - v\_2 , ….  $ where $v\_t$ is sampled from the binomial in e.q. (7).
>
> To illustrate, referring to Figures 1.a and 1.c, the generation of the single community in level $l=1$ is associated with a total of 29 edges/events, determined by the parent node of this community. Consequently, the edge probabilities of this community follow a Multinomial distribution. However, we model it in an autoregressive (AR) manner as a sequence of Binomials and Multinomials, as outlined in Theorem 3.3.
>
> **Questions:**
>
> 2, 3) Thank you for raising this analysis. Since the structure of the subgraph evolves during recursive community generation, the graph-level features are not similar during the community generation.  Therefore, to address this issue, we have concatenated the edge features with graph-level features $pool( \mathbf{h}\_{\hat{\mathcal{C}}^{l}\_{i,t} }) $ , (the term that was also used to model the probability of total weights $v\_t$ in eq. (7) ). Consequently, the self-edges' outputs primarily depend on graph-level features, ensuring that their probabilities are not similar.  Moreover, we observed that the model was able to generate samples, with heterogeneous communities with varying sizes.
>
> 4 ) Thank you for your point. In the context of generating graphs in an autoregressive (AR) manner, there are two primary approaches: I) Generating a graph by adding one edge at a time (edge AR), exemplified by methods like GraphRNN. II) Generating one node and its corresponding group of edges at a time (node AR), as demonstrated by models like GRAN.
> Lemma 3.2 provides a way to generate edges recursively, making it suitable for modeling a multinomial distribution in AR models such as GraphRNN. However, this approach requires a high number of generation steps, approximately $O(n^2)$, equivalent to completing the adjacency matrix element by element.
>
> On the other hand, Theorem 3.3 permits grouping the edges of each node and generating a community in a node-by-node fashion. This means generating a group of edges corresponding to a row of the adjacency matrix at each step (as depicted in Figure 1.c), resulting in significantly faster generation. For this reason, we adopted Theorem 3.3 to model the probability of edges in each community instead of using Lemma 3.2.

---

> > ### Comment · Reviewer_1z1k · 2023-08-20
> > **Upscore decision**
> >
> > I appreciate the authors for providing thorough responses to my concerns, I am glad to see that most of them are addressed. Hence I am biased towards accepting the paper if these explanations can be properly incorporated in the revision.

---

> > > ### Author Response · Authors · 2023-08-21
> > >
> > > We sincerely appreciate your valuable feedback and are grateful for your recognition of our efforts to address your concerns. In our initial submission, due to the the page limit constraints, we focused on key aspects and experimental results to effectively convey the core concepts and contributions of our work. However, with the provision of additional space in the final version, we will integrate the supplementary analysis, comprehensive clarifications and new results that directly respond to your concerns, as well as those that emerged and have been thoroughly discussed during the rebuttal phase.

---

### Author Rebuttal · Authors · 2023-08-10

Thank you to the reviewers for their valuable comments and analyses, which have contributed to the clarity of the paper. We have addressed the questions and concerns of each reviewer individually and in the specified order. Your feedback will be helpful in enhancing the quality of our work.

## statistics of the graph datasets
Here we summarize some statistics of the graph datasets

dataset | $max(n)$  | $avg(n)$  | $avg(\|c\|\_{max})$  | $avg(n\_c)$ | $avg(modularity\_{gen})$ | $avg(modularity\_{test})$
---|---|---|---|---|---|---
Enzyme | 125 | 33 | 9.8 | 4.62 | 0.62 | 0.59
Ego | 399 | 144 | 37.52 | 8.88 | 0.66 | 0.56
Protein | 500 | 258 | 26.05 | 13.62 | 0.8 | 0.77
SBM | 180 | 105 | 31.65 | 3.4 | 0.59 | 0.6
3D point Cloud | 5K | 1.4K | 97.67 | 18.67 | 0.88 | 0.85

Where $\|c\|\_{max}$ denotes the maximum size of clusters and  $n\_c$ is the number of clusters in each graph .

---

> ### Author Response · Authors · 2023-08-10
> **Loss Function**
>
> **Loss Function:**
> According to equations (1) and (2), the log-likelihood for a graph sample can be written as
> $$ \\log p(\\mathcal{G}^) = \sum\_{l=1}^{L} (\sum\_{i} \\log p(\mathcal{C}\_{i}^l  | \mathcal{G}^{l-1}) \times \sum\_{i,j}  \\log p(\mathcal{B}\_{ij}^l  | \mathcal{G}^{l-1}, \\{ \mathcal{C}\_{i} \forall \mathcal{C}\_{i} \in \mathcal{G}^{l} \\}) )$$
>
> Therefore, one need to compute the log-likelihood of the communities and cross-community components. The log-likelihood of the cross-communities are straightforward using eq. (8).
> For  the log-likelihood of the cross-communities, as we break it to subsets of edges for each node in AR manner, with probability of each sunset modeled in eq. (6), the log-likelihood is reduced to
> $$ \\log p(\\mathcal{C}\_\{i\}^l  | \\mathcal{G}^{l-1}) = \\sum\_{t=1}^{ \| \\mathcal{C}^{l} \|} \\log p(u\_t (\\hat{\\mathcal{C}}^{l}\_{j}) )$$
> where $p(u\_t (\\hat{\\mathcal{C}}^{l}\_{j}) )$  is defined  in eq. (6) as a mixture of product of binomial and multinomial.
>
> For training, we randomly sample $s$ subgraphs of the communities at each level so given  $s\_i$ subgraphs are from community $\\mathcal{C}\_\{i\}^l $ then, we estimate the conditional generative probability for this community by averaging the loss function over all the subgraphs in that community multiplied by the size of the cluster:
> $$ \\log p(\\mathcal{C}\_\{i\}^l  | \\mathcal{G}^{l-1}) = \| \\mathcal{C}^{l} \| * mean( [  \\log p(u\_t (\\hat{\\mathcal{C}}^{l}\_{t} )  \~ \\forall \~ t \\in s\_i ] )$$
>
> Since the Binomial and Multinomial are from the exponential family distribution, their log-likelihood reduces to Bregman divergence [1]. The binomial log likelihood is a general form of Bernoulli log likelihood (binary cross entropy)
> , and multinomial log likelihood is a general form of multinoulli (categorical) log likelihood (multinoulli cross entropy).
>
> [1] Banerjee, Arindam, et al. "Clustering with Bregman divergences." Journal of machine learning research 6.10 (2005).

---

> > ### Author Response · Authors · 2023-08-16
> > **On Model sizes and Sampling Speed Comparison**
> >
> > **Model size:**
> >
> > The following table displays the parameter counts of HiGen across different datasets.
> > To ensure the HiGen models of the same size or smaller size than GRAN, the main baseline of our study, we conducted the experiments for SBM and Enzyme datasets with reduced sizes. The resulting model sizes and performance metrics are presented below. For the SBM dataset, we set the hidden dimension to 64, and for the Enzyme dataset, it was set to 32.
> >
> > |Model | Protein | 3D Point Cloud | Ego | Enzyme | SBM |
> > |--- | :---: | :---------: | :---: | :---: | :---|
> > |**GRAN** 	  | 1.75e+7  | 5.7e+6 | 1.5e+7  | 1.54e+6 | 3.16e+6 |
> > |**HiGen** 	| 4.00e+6 | 6.26e+6 | 3.96e+6 | 1.48e+6 | 3.19e+6|
> >
> >
> > Comparison of generation metrics on Enzyme. The hidden dimension of HiGen is set  to 32.
> >
> >  Model 	|  Deg. $\downarrow$ | Clus. $\downarrow$ | Orbit  $\downarrow$
> > ---|---|---|---
> > Training set  |  0.0011 | 0.0025 | 3.7e-4
> >  **GraphRNN**  	| 0.017 | 0.062 | 0.046
> >  **GRAN**   	   | 0.054 | 0.087 | 0.033
> >  **GDSS**   	   | 0.026 | **0.061** | 0.009
> >  **HiGen** 	   | **6.8e-3** | 0.067 | **1.3 e-3**
> >
> > Comparison of generation metrics on Stochastic Block Model. The hidden dimension of HiGen is set  to 64.
> >
> >  Model     |  Deg. $\downarrow$  | Clus. $\downarrow$  | Orbit$\downarrow$  | Spec.\,$\downarrow$
> > ---|---|---|---|---
> >  Training set   | 0.0008  | 0.0332  | 0.0255  | 0.0063
> >  GraphRNN      | 0.0055  | 0.0584  | 0.0785  | 0.0065
> >  GRAN		  | 0.0113  | 0.0553  | 0.0540  | 0.0054
> >  SPECTRE 	  | 0.0015  | 0.0521  | 0.0412  | 0.0056
> >  DiGress 	  | **0.0013** | **0.0498**   | 0.0433 | -
> >  HiGen   	  | 0.0015  | 0.0520  | **0.0370**  | **0.0049**
> >
> >
> > For these experiments, the hidden dimensions were set to 64 for the Protein, Ego, and Point Cloud datasets, and the Stochastic Block Model and 32 forEnzyme datasets. In contrast, the GRAN models utilized GNN layers with hidden dimensions of 128 for the Stochastic Block Model, Ego, and Enzyme datasets, 256 for the Point Cloud dataset, and 512 for the Protein dataset.
> >
> > These tables clearly indicate that despite smaller or equal model sizes, HiGen outperforms GRAN's performance.
> > This emphasizes the efficacy of hierarchically modeling communities and cross-community interactions as distinct entities.
> > This can be explained by the fact that the proposed model needs to learn smaller community sizes compared to GRAN, allowing for the utilization of more compact models.
> >
> > **Sampling Speed Comparison:**
> >
> > In the table below, we present a comparison of the sampling times between the proposed method and its primary counterpart, GRAN, measured in seconds. The sampling processes were carried out on a server machine equipped with a 32-core AMD Rome 7532 CPU and 128 GB of RAM.
> >
> > | Model 	 | Protein |   Ego   |   SBM   | Enzyme |
> > |------------|---------|---------|---------|--------|
> > | **GRAN**  	 |  46.04  |  2.145  | 1.5873  | 0.2475 |
> > | **HiGen**|   1.33  |  0.528  | 0.4653  | 0.1452 |
> >
> > From the table, it is evident that HiGen demonstrates a significantly faster sampling, particularly for larger graph samples.

---

### Decision · Program_Chairs · 2023-09-21

**Decision:**

Reject

**Comment:**

The paper received mixed borderline ratings. The reviewers like the proposed model and its potential in addressing large-scale graph generation, but also share major concerns regarding its unclear model description, insufficient model evaluation and discussion with related works. The authors provided several additional results and clarified some misinterpretations in the rebuttal. Three reviewers then increased their ratings and leaned towards the positive.

 However, during discussion among reviewers and ACs, the majority of the reviewers, while valuing the proposed idea, still share concerns regarding the model evaluation and paper’s clarity. The ACs concur with the reviewers and believe that substantial improvements are necessary before publication. Specifically, the paper needs to enhance clarity in its contribution, model description (e.g., theorem 3.1) and limitations (e.g., potential overfitting issues). Additionally, it should include necessary technical and experimental details as suggested by the reviewers. The paper can also be strengthened by including in-depth discussions or comparisons with the relevant hierarchical baselines or graph generative baselines as highlighted by the reviewers. Furthermore, evaluations on larger benchmarks with a deeper hierarchy (e.g., L>2; results on 3D point cloud dataset in the appendix can be a good starting point ) would better demonstrate the potential of the proposed model. As such, the ACs would recommend rejecting the paper in its current form.  The paper would benefit from a thorough revision and another round of reviews before being considered for acceptance.